# Reaction time coupling in a joint stimulus-response task: A matter of functional actions or likable agents?

**Zoe Schielen**[☉], **Julia Verhaegh**[☉], **Chris Dijkerman, Marnix Naber**[iD]*

Experimental Psychology, Helmholtz Institute, Faculty of Social and Behavioral Sciences, Utrecht University, Utrecht, The Netherlands

☉ These authors contributed equally to this work.
* marnixnaber@gmail.com

## Abstract

Shaping one owns actions by observing others' actions is driven by the deep-rooted mechanism of perception-action coupling. It typically occurs automatically, expressed as for example the unintentional synchronization of reaction times in interactive games. Theories on perception-action coupling highlight its benefits such as the joint coordination of actions to cooperatively perform tasks properly, the learning of novel actions from others, and the bonding with likable others. However, such functional aspects and how they shape perception-action coupling have never been compared quantitatively. Here we tested a total of hundred-fifteen participants that played a stimulus-response task while, in parallel, they observed videos of agents that played the exact same task several milliseconds in advance. We compared to what degree the reaction times of actions of agents, who varied their behavior in terms of functionality and likability in preceding prisoner dilemma games and quizzes, shape the reaction times of human test participants. To manipulate functionality and likability, we varied the predictability of cooperative behavior and correctness of actions of agents, respectively, resulting in likable (cooperative), dislikable (uncooperative), functional (correct actions), and dysfunctional (incorrect actions) agents. The results of three experiments showed that the participants' reaction times correlated most with the reaction times of agents that expressed functional behavior. However, the likability of agents had no effects on reaction time correlations. These findings suggest that, at least in the current computer task, participants are more likely to adopt the timing of actions from people that perform correct actions than from people that they like.

## 1. Introduction

The tendency to adjust one's own actions to the perceived actions of others can be found in a variety of behavioral phenomena. For example, imitation and mimicry are frequently observed forms of behaviors and involve the copying of posture, speech, and other types of muscle movements during interaction. The mutual copying of behavior by people is typically

**Competing interests:** The authors have declared that no competing interests exist.

attributed to automatic perception-action coupling. The observation of another person's action automatically influences the observer's mental representation of the action, which overlaps with the observer's own, currently planned or active action representation and scheme [1–3]. It is important to note that several forms of imitation exist, including reproduction attempts of the spatial movement patterns of a person's actions (e.g., yawning), time-coupling of (repetitive) behavior (e.g., synchronization during dancing), and becoming as ready as another person to quickly perform actions (e.g., reaction time (RT) coupling). It is the latter form that has received little scientific attention and will be of interest to the current study.

Most evidence that the perception of other's actions influences the readiness to respond and thus the reaction time of an observer comes from studies using joint action stimulus-compatibility tasks (for a review, see [4]). A popular perspective on such effects is that the response times to stimulus onsets, as typically measured during such tasks, depend on the mere presence of another person, likely caused by the automatic sharing of and interference between mental representations of the actions of others (for a review, see [5]). The crossover of perception-action schemes between people could be considered as a mere side-effect of how the human cognitive system is structured. Only few have considered why the perception-action system is structured the way it is from an evolutionary or functional perspective. For example, some suggest that perception-action coupling–whether it is the spatial copying of actions, temporal synchronization, or RT coupling–is a crucial behavioral mechanism for everyday life as it allows (young) people to learn useful skills from others [6–11]. More recent literature on imitation and behavioral synchronization explains the underlying function as a social bonding and understanding mechanism [12–18]. Similarly, the time-coupling and synchronization of behavior during joint action strongly depends on the social context and a person's goal to cooperate [5, 15, 19–24].

An alternative, non-social explanation for behavior coupling (and synchronization) also exists. Termed as the "Referential Coding" account of coupling behavior in joint action tasks (i.e., mostly in the joint Simon task), Dolk and colleagues [4, 25] suggest that an action is stored in the brain as a code (representation) consisting of multiple features, such as the speed of a response (e.g., fast or slow) and the physical changes it evokes in the environment (e.g., the clicking sound of a button press). The features of self-produced and observed actions, independent of whether these are produced by a social being, can overlap in code content. When a feature of two actions (e.g., the location of a button) becomes incongruent, interference in coding occurs, which consequentially may slow reaction times. The strongest evidence for this non-social account of behavioral coupling comes from studies showing that reaction times in joint action tasks do not depend on the social identity of co-acting agents but mostly on how participants process spatial aspects of actions of others and how such observed action representations interfere with one's own action plans [26, 27]. Also, reaction time interference or synchronization even emerges in competitive, non-social contexts [28, 29]. In the light of the aforementioned social and non-social accounts, we here investigate to what degree social and non-social aspects are relevant to the coupling of reaction times of action responses to stimuli.

An interesting approach to study the importance of social versus non-social effects on RT coupling is the use of virtual agents because their behavior and the social context can be controlled well. Interestingly, virtual co-actors can even be invisible and still influence reaction times of participants in joint stimulus-compatibility tasks, though only when participants believe that the agent is a biological being rather than digital entity [30]. It is to be noted that the digital or virtual aspect of an agent's presence may reduce the amount of attention for their actions and consequentially temper effects on reaction time coupling as compared with a "live" agent presence which naturally attracts more attention [31]. Thus, besides the belief of having an intentional (virtual) partner (also see [32]), sufficient attention for the partner's

actions, either drawn bottom-up through salient events [25, 33] or top-down though instruction [34, 35], is required to modulate reaction times of the observer. Similar arguments about the importance of the focus of attention have been provided by studies on mimicry of physiological states, such as pupil size [36, 37]. Social aspects may alter the amount of attention for actions and reaction times indirectly, but social aspects do not directly determine the degree of reaction times coupling, mimicry, or imitation.

With the current study we address to what degree the time-coupling of responses to stimuli can be affected by a virtual agent. We do this in the context of the aforementioned functional accounts of behavioral coupling and synchronization. Remarkably, these theories, concerning learning, social, and cooperative views on the functional benefits of imitative perception-action coupling, have never been compared. We argue that, depending on the circumstances of the task, if one type of benefit is more relevant than the other, perception-action coupling probabilities should increase when the factors causing that benefit become more obvious. Here we examine how strong the factors of functionality and likability affect the coupling of reaction times between a participant and a virtual agent. Will it be mostly the usability of observed actions and the possibility to learn functional behavior from others, or mostly the likability of the observed person and the chance to bond, or both equally?

Before we continue to the outline of the methods, it is first necessary to discuss the studies that investigated the factors of functionality and likability in isolation. Both factors are known to affect, for example, the imitation of bodily and virtual actions. The observation of a functional action as compared to dysfunctional action increases the probability that a subsequently performed action is imitated [38]. Also, children already at the age of 18 months only imitate actions after they determined that the action was intentional and thus functional [39]. Imitation may thus facilitate or is inherent to becoming acquainted with useful skills. However, benefitting from the observation of useful actions is not sufficient to explain all findings in the literature. For example, although the observation of dysfunctional actions less likely affects a person's own actions [38], diverging forms of imitation remain to occur even when the observed behavior is rather strange [40], has no function [41], has negative effects on the mimicker [29], or is discouraged with monetary incentives [42]. This raises the question whether it is also the innate, social goal of humans to connect with others that makes them incorporate dysfunctional behaviors of others into their system? The most evident finding that supports this account of imitation, perception-action coupling, and behavioral synchronization is that likable persons are more often imitated [43], people are more likely to synchronize behavior with likable person [19], and persons that imitate are liked more [44, 45], which enhances their individual advantage through team efforts [46]. However, the factor of likability does not sufficiently explain all forms of such imitative behaviors, because rather strong instances of RT coupling and other imitative forms have been observed in nonsocial and competitive situations [28, 29, 42, 47]. The question thus remains: how relevant are functionality and likability for RT coupling to occur?

To assess the relative contributions of functionality and likability to the tendency to couple reaction times, we conducted three experiments in which human "followers" that played a stimulus-response compatibility task in parallel with and slightly behind agents (virtual confederates) that functioned as "Leaders" (for an example of other leader-follower tasks, see [48]). The agents' behaviors were manipulated in preceding games to come across as likable, dislikable, functional, or dysfunctional. The synchronization of the timing of actions will be measured as the correlation between the reaction times (RT) of the participants and observed agents.

## 2. Experiment 1

### 2.1 Introduction

The first experiment examined the effect of the likability and functionality manipulations. In this experiment the factors functionality and likability were individually manipulated per agent.

### 2.2 Methods

**2.2.1 Participants.** Thirty individuals (age: M = 24.4, SD = 9.89, range: 18-55; 22 women; 2 lefthanded) participated in the first experiment. Participants received study credit for participation. All participants had normal or corrected to normal vision, gave their informed written consent before participation, were naive to the purpose of the experiment, and were debriefed about the purpose afterwards. The current study is compliant with the ethical principles of the Declaration of Helsinki and was approved by the local FETC ethics review board. The experiment lasted approximately 45 minutes.

**2.2.2 Apparatus.** Stimuli were shown on a desktop Dell computer (Dell, Round Rock, TX, USA), operating on Windows 7 (Microsoft, Redmond, WA, USA) and Matlab version r2010a (Mathworks, Natick, MA, USA). The desktop computer screen had a resolution of 1920x1080 pixels and a refresh rate of 60 Hz.

**2.2.3 Experimental design, procedure, and stimuli.** Experiment 1 consisted of four parts. In the first part, participants were introduced to an opponent. In the second part, participants played two computer games (Fig 1; see S1 Video for an example of a single trial of each game). Participants alternated between the games in blocks of several trials. Participants played these games with agents virtually, and they were led to believe, through verbal and textual explanations, that the agents were real persons whom were seated behind a computer in other rooms and that the hand movements of the agents were livestreamed to the participants. One game consisted of the prisoner's dilemma game (PD) to manipulate to what degree agents (i.e., opponents) cooperated, therewith influencing the degree they were (dis)liked. Another game was a color action game (CA), which had two goals: (1) to manipulate whether agents played good (i.e., functional) or bad (dysfunctional), and (2) to measure the degree of similarity in reaction times (i.e., a correlation; see Naber et al., 2013, for more information about this operationalization) between participants and agents as a proxy of perception-action pairing. The third part of experiment 1 consisted of a short questionnaire to confirm that the agent manipulations had the expected effects on likability and functionality as perceived by the participants.

*2.2.3.1 Agents.* Participants played the PD and CA games with one agent per block. Before each block of 20 trials (a trial consisted of one action during the prisoner's dilemma game and actions during the action game; top panel in Fig 1B), participants were shown the agent's name and a picture of his or her face (Chicago face database; [49]) on the screen to notify them that they would start playing with a new agent. The agents consisted of five females and one male and expressed neutral emotions. The reason for this female-male distribution was to match the participants group gender ratio, based on the psychology students in the Netherlands. Each agent varied in the degree of behaving cooperatively during the prisoner's dilemma game and functionally during the color action game (see Table 1). For each participant, the behavioral characteristics of the agents were randomly coupled to the set of names and faces. Two of the six agents served as a baseline and behaved functionally and cooperatively at an intermediate level.

*2.2.3.2 Likability manipulation—prisoners dilemma game (PD).* The PD game is a socio-economic game during which two participants need to cooperate to achieve the best test results for both of them [50]. The PD is often used to investigate how a participant's decision-making

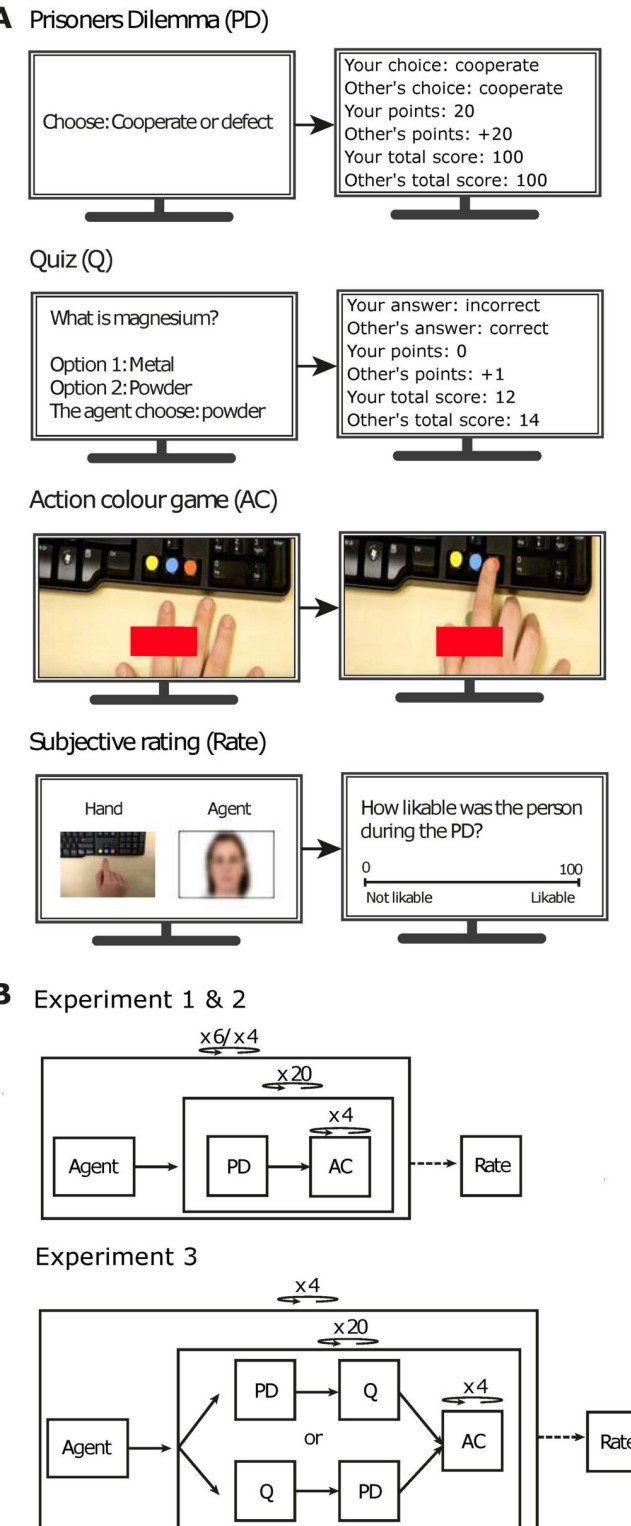

**Fig 1. Stimuli and procedures.** The figures represent a reconstruction of the displays shown to the participants during the games. In the prisoner's dilemma game (PD; top panel in A) the participant chose between cooperate and defect, and then obtained an overview of the agents choice and how this affected their rewards. In experiment 3, participants also participated in a quiz (Q; second panel in A), choosing an answer from two answer options per question, and knowing the agent's answer in advance. In the action color game (AC; third panel in A) participants saw the video of

an agent pressing a colored button and were ought to press a button with a color corresponding to the cue consisting of colored rectangle. At the end of the experiment participants rated the agents on their likability and functionality (Rate; bottom panel in A). Each of these parts took place in an order as displayed in (B). Participants played 80 games with each agent within a block.

is affected when playing against a cooperative versus uncooperative agent. We here exploit the PD because it has shown to be highly effective in evoking emotions of frustration, annoyance, and strong dislike expressed towards uncooperative agents by participants [51]. Vice versa, the perceived likability of a virtual agent influences the willingness to cooperate by the participant [52, 53]. During the PD game, a participant played a computer game with an agent with the goal to collect as many points as possible (top panel in Fig 1A). Per trial, the participant and the agent decide whether they want to cooperate or defect. When both cooperate, both get points. When one cooperates and the other defects, the cooperative player loses points and the other gains many points. When both defect, both loose points. The points scored based on the combination of choices of both the participant and the agents can be found in **S1 Table**. An overview of the agent's choice during the PD game and the accumulation of points that the participant and agent received were displayed after each PD trial (See Fig 1, top panel in A). Each agent was programmed with a different ratio of cooperate versus defect choices (number of cooperative trials: 16/20 for likable agents; 10/20 for neutral agents, 4/20 for dislikable agents).

*2.2.3.3 Functionality manipulation–color action game (CA)*. Four actions during the CA game followed directly after each prisoner's dilemma action. This game achieved two goals in experiment 1: (1) manipulate the functionality of actions of observed agents, and (2) measure the degree of perception-action coupling. Our game was inspired by a finger-tapping task in which participants observe another person making an action with their finger and are then cued to perform an action as well [54]. The participant's task was to respond to a visual cue (a colored rectangle: yellow, blue, red) as fast as possible by moving a finger from a starting point towards one of three buttons with a color corresponding to the cue (third panel in Fig 1A). Crucially, around the time the cue appeared, a pre-recorded video of the agent's hand appeared, moving his or her finger to one of the three buttons (see S1 Video). The CA game, essentially consisting of a virtual form of a joint compatibility task, incorporated cue-congruent and cue-incongruent actions. Cue-congruent trials consisted of trials in which the chosen color by the agent was in line with the shown color of the cue. In cue-incongruent trials the color choice of the agent differed from the cue's color. Note, however, that any effects related to compatibility fall out of scope of the current study. The main purpose of the correct and incorrect agent trials in the CA game was to manipulate action functionality but not to study compatibility effects.

We told the participants that this was a live video stream of the partner that they played with in the preceding PD game. To develop these pre-recorded agent videos, a separate group

**Table 1. Behavioral characteristics per agent in experiment 1.**

| Agent type | Prisoner's dilemma game *Percent cooperative trials* | Color action game *Percent correct choices* |
|---|---|---|
| Functional | 50% | 80% |
| Dysfunctional | 50% | 20% |
| Likable | 80% | 50% |
| Dislikable | 20% | 50% |
| Neutral (2x) | 50% | 50% |

of six participants had been previously invited to perform a color cue reaction task with 80 trials. During the color action game of the actual experiment, the agents varied in the ratio of cue-congruent and cue-incongruent trials (number of correct trials: 64/80 for functional agents; 40/80 for neutral agents; 16/80 trials for dysfunctional agents). The movement onsets in these agent videos were controlled at a temporal resolution of milliseconds with a video editing program. The range of an agent's movement onset latencies was shifted to vary -200ms to +100ms around the cue onset for the participant, while maintaining the original distribution of latencies. This range ensured that it was highly likely that the agent started moving in the video before the participants started moving. This timing aspect was crucial because the participant's pre-observation of the agent's action was expected to influence their reaction time through (automatic) perception-action coupling. The cue was superimposed on top of the agent's hand in the video to ensure that the participant's focus of attention was drawn to the agent's action. We here define the agent's reaction time as the duration between the cue onset for the participant and the movement onset of the agent, and we define the participant's reaction time as the duration between the cue onset for the participant and the button press of the participant. As operationalized before in Naber et al. (2013), the degree of perception-action coupling is calculated as the correlation between the reaction times of the agent and participant across CA games.

After pressing a colored button, the participant received feedback about their choice and reaction times. A small dot was shown on the location of the previous cue with a color corresponding to a specific feedback type: green if the correct color was chosen, magenta for an incorrect choice, and cyan when participants responded too late (>750ms after the cue).

*2.2.3.4 Subjective ratings*. After playing all trials, we verified that the variations in behavioral characteristics across agents were noticed by the participants. We asked them to rate each agent on a functionality (i.e., How functional/well did the agent made his/her choices in the color action game?) and likability scale (i.e. how likable the agent during the prisoner's dilemma was game) that ranged from 0 to 100 percent (fourth panel in Fig 1A; see S2 Table for all the subjective rating questions). The face, name, and a snapshot of the video with the hand of the rated agent was shown together with questions.

**2.2.4 Analysis.** Trials during which the participants responded too late were excluded from the analysis. Both incorrect trials (i.e., participants selected the wrong color) and correct trials were used for RT coupling. We first investigated the presence of RT coupling by calculating the slopes of linear regression fits to RTs of agents versus participants. We operationalized perception-action coupling as Pearson's *r* correlation coefficient between reaction times of participants and agents of all but except late trials of the CA game. Note that we treat perception-action coupling as a broad construct, not allowing to dissociate between different forms, such as conscious reaction time synchronization or unconscious mimicry (see Discussion for more information). Due to the scope of our research question and the relatively small number of trials, we ignored effects of stimulus-response compatibility and thus restricted the analysis to variations in the RT correlations across agent types.

The slope of the linear fits per individual were compared to zero with t-test comparisons. The correlation coefficients were log normalized for the statistical analyses. We performed a one-way repeated measures ANOVA with the factor *agent type* as independent variable, and *RT coupling* as dependent variable. Degrees of freedom were corrected with the Huynh-Feldt-Lecoutre method in case of a violation of sphericity. Post-hoc testing was performed with two-tailed, paired student's t-tests to assess the difference in RT coupling between functional and dysfunctional, and likable and dislikable agents. While we manipulated the agent's cooperativeness in the prisoner's dilemma game, we referred to this manipulation as likability, a consequence of cooperativeness. To confirm that participants noticed the likability and

functionality differences across agents–the awareness of an agent's actions is important for RT coupling to occur (Naber et al., 2013)–we performed two two-way repeated measure ANOVAs with the factors *agent likability* and *agent functionality* as independent variables, and *subjective ratings* on likability and functionality respectively as dependent variables.

## 2.3 Results & discussion

**2.3.1 Response accuracies.** The percentage of trials in which participants responded too late (M = 5.4%, SD = 3.9%) or chose a wrong, not cued color (M = 4.0%, SD = 3.4%) were small, indicating that participants performed the CA games well. These percentages did not differ across agent types (Late trials: $F(4, 116) = 0.70$, $p = 0.591$, $\eta_p^2 = 0.02$; Incorrect trials: $F(4, 116) = 1.03$, $p = 0.395$, $\eta_p^2 = 0.03$), indicating that likability or functionality manipulations did not affect the accuracy of a participant's actions.

**2.3.2 Coupling of RTs between agents and participants.** As explained in the Methods, we assumed that reaction times of the participants would be coupled to the reaction times of agents across trials if RT coupling took place. To confirm the correctness of this assumption, we first examined the reaction times of the participants as a function of the reaction times of the agents with linear regression fits per participant (S1A Fig). A positive slope of the linear regression line was found across all participants (i.e., significantly different from zero: $t(29) = 14.72$, $p < .001$, Cohen's $d = 2.69$), indicating that when an agent reacted relatively fast or slow in a trial, the participant also reacted faster or slower, respectively. The slope of a linear fit indicated an average increase in reaction time of 0.27ms (SD = 0.10ms) across participants per 1ms increase in reaction time by the agent. The correlation between RTs of agents and participants was weak (M = .20, SD = .06), but the population of correlations across participants differed significantly from zero ($t(29) = 17.71$, $p < .001$, $d = 3.23$). In sum, the fact that the participants' reaction times varied in the same direction as the agents' reaction times suggests the manifestation of perception-action coupling.

**2.3.3 Influence of likability and functionality on RT coupling.** Next we established whether the degree of pairing changed as a function of the functionality and likability of the agents. The pattern of correlations across the likability and functionality manipulations in valence (i.e., positive, neutral, or negative) suggested that especially differences in functional behavior across agents affected RT coupling (Fig 2A; see S2A Fig for an alternative representation of data). The correlations significantly differed across the five agent types ($F(4, 116) = 5.15$, $p = .001$, $\eta_p^2 = 0.15$). Functional versus dysfunctional agents evoked significantly different degrees of coupling ($t(29) = 4.30$, $p < .001$, $d = 0.78$) while likable versus dislikable agents did not ($t(29) = 1.22$, $p = .233$, $d = 0.22$). This indicated that participants were more affected by the actions of a functional agent than a dysfunctional agent, but that did not apply to a likable agent as compared to a dislikable agent. To investigate the differences in degree of RT coupling between the likable versus functional agent characteristic, the effect of functional versus dysfunctional agents was compared with the effect of the likable versus non-likable agent. We observed no difference in the coupling effect of functional versus dysfunctional and likable versus dislikable agents ($t(29) = 1.74$, $p = .093$, $d = 0.32$). However, previous findings showed that the probability of pairing functional actions decreases as time passes (Naber et al., 2016). A split-trial analysis confirmed this and revealed that the effect of functionality as compared to likability on RT coupling was significantly stronger in the first two of four color action game trials ($t(29) = 2.26$, $p = .031$, $d = 0.41$) but not the last two trials ($t(29) = 0.80$, $p = .431$, $d = 0.15$). As such, at this point we can conclude three things: (1) agents that made few errors during the color action game affected more strongly the participants' actions than agents that made many errors, (2) cooperative and likable agents did *not* affect participants more than

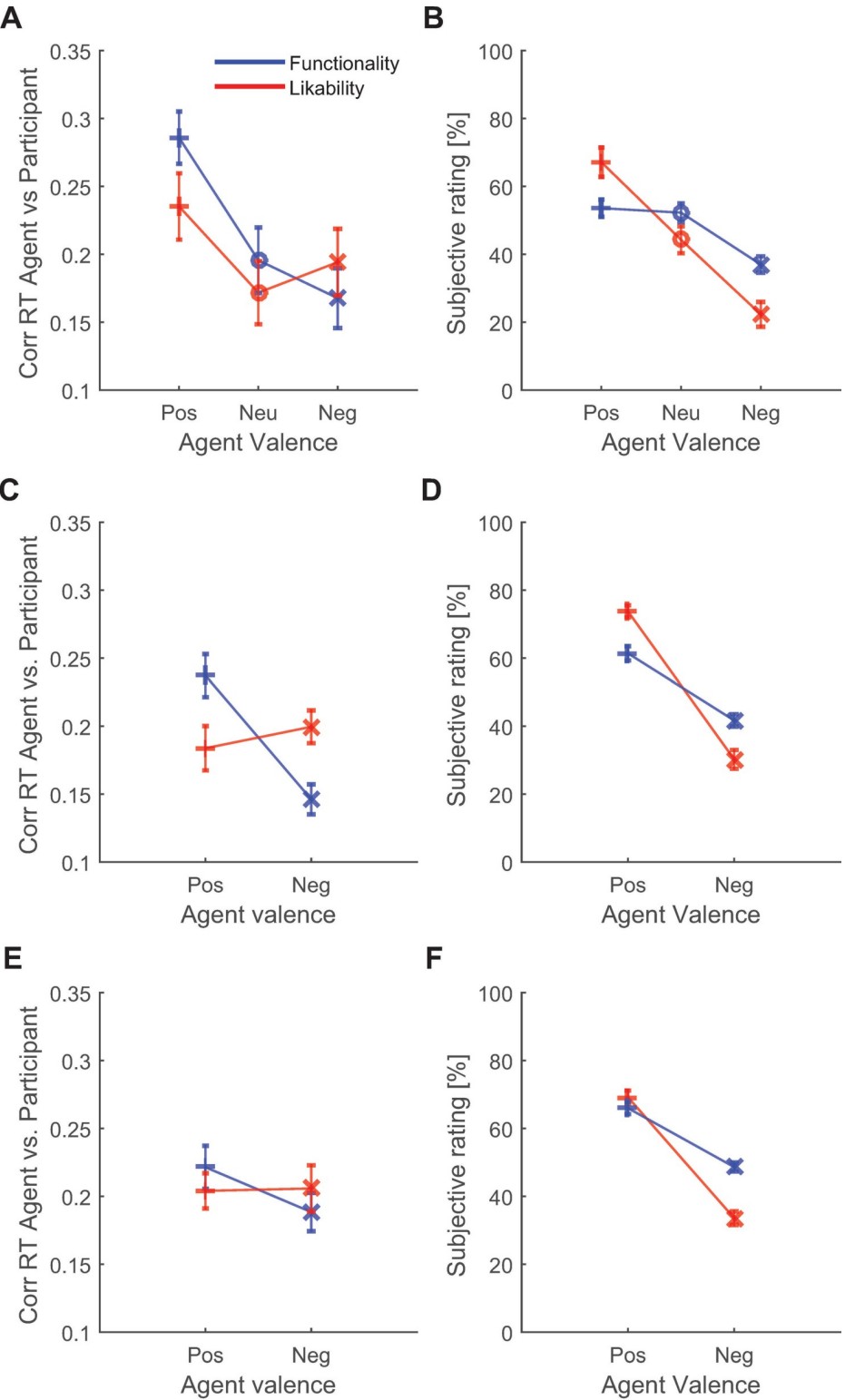

**Fig 2. Experiment 1 results.** The correlations (Corr) of reaction times (RT) between agents and participants were largest when participants played with a functional (blue line; Pos = positive) agent and were weakest when participants played with a dysfunctional agent in experiment 1 (Neg = negative; Neu = neutral) agent (A). We observed no difference between likable and dislikable agents (red line). Subjective ratings by participants confirmed that they were aware of the differences in functionality and likability across agents (B). Similar patterns of correlations (A) and subjective ratings (B) across agents were present in experiment 2 (C, D) and experiment 3 (E, F).

agents that were uncooperative and dislikable, and (3) the difference in the strength of RT coupling between a functional and dysfunctional agent is greater than the difference between a likable and dislikable agent.

**2.3.4 Possible alternative RT strategies.**   One possible confound that needs to be addressed is that the variations in RT correlations could have been a consequence of an employed waiting strategy by participants rather than a consequence of perception-action coupling. In other words, when the participants learned after several trials that an agent almost made no mistakes, they may have waited until a functional agent responded before initiating a response themselves. This will automatically lead to apparent reaction time correlations. Although this would still be the result of a perception-action sequence, it would not be the result of incorporating a representation of another person's actions. We inspected this possibility by checking for overall *slower* RTs of participants when observing functional than dysfunctional agents as indication of the employment of a waiting strategy. However, functional agents evoked *faster* RTs in participants (M = 505ms, SD = 58ms) than dysfunctional agents (M = 529ms, SD = 59ms; $t(29) = 3.45$, $p = .002$, $d = 0.63$) and no difference was seen in RTs between likable (M = 520ms, SD = 57ms) and dislikable agents (M = 518ms, SD = 58ms; $t(29) = 0.25$, $p = .805$, $d = 0.05$). This means that participants responded 24ms faster when playing with a functional agent, likely reflecting stronger perception-action coupling as they were following the agent's responses rather than postponing their responses until the agent made a choice. Moreover, this difference was so small that it is unlikely the result of a deliberate strategy.

**2.3.5 Subjective awareness of agent manipulations.**   The finding of no significant difference in pairing effects between liked and disliked agents could have been the result of a failed manipulation of agent likability. To check this, we examined whether the conditional pay-off matrix in combination with the participant's choices produced the expected difference in points for participants playing against an uncooperative versus cooperative agent. Indeed, the likability manipulation successfully affected the points scored by participants (likability: $F(1.4, 41.7) = 2013.55$, $p < .001$, $\eta_p^2 = 0.99$; Positive: M = 187.33, SD = 42.75; Neutral: M = -47.67, SD = 27.00; Negative: M = -265.67, SD = 11.94), indicating that the agent likability manipulation was successful on the level of the participants' scores.

An alternative explanation for the lacking effect of agent likability on perception-action coupling is that participants were not aware of the variations in cooperative behavior across agents, despite the divergent patterns of changes in scores across agents. However, the differences in the participant's subjective likability ratings across agents indicated that they were aware of this (Fig 2B; for statistics, see S3 and S4 Tables). The same applied to the functionality ratings, that is, participants rated functional agents as better players than dysfunctional agents. In fact, the difference in ratings between the liked (positive valence) and disliked (negative valence) agent (M = 44.8, SD = 36.5) was significantly bigger than between the functional and dysfunctional agent (M = 16.6, SD = 11.9, $t(29) = 4.80$, $p < .001$, $d = 0.88$). This means that the manipulation in agent cooperation during the PD game had a stronger effect on the perceived likability than the manipulation in agent error rates during the CA game had on the perceived functionality. The frequent verbal expressions like "Come on!", "Argh!", and "Boo!" made by participants during the prisoner's dilemma game, and conversations with the participants after the experiment further confirmed that participants were considerably annoyed by the uncooperative agents.

Although the likability manipulation successfully changed the opinions of participants, playing against a disliked agent had no monetary consequences. Even so, the likability manipulation did not result in differences in reaction time correlations. In experiment 2 we addressed this by putting more emotional weight on a trial in which an agent defected a deal: (1) we

rewarded (or punished) the participants with monetary incentives, (2) we changed the pay-off matrix such that a defect punishment had relatively more impact on the scores than in experiment 1, and (3) to simplify the likability (and functionality) manipulations and to make the behavioral differences across agents more conspicuous in experiment 2, we combined the manipulations within four instead of six agent types.

## 3. Experiment 2

### 3.1 Introduction

Experiment 2 builds on experiment 1, with the aim to provoke a higher social involvement of the participant by strengthening the effect of the likability manipulation.

### 3.2 Methods

All methodological aspects of experiment 2 were similar to experiment 1, except for the group of participants, the experimental design, and the statistical design. The group of participants in the second experiment consisted of 41 people (M = 21.63, SD = 1.68, range: 19–28, 30 women; 6 left handed). To increase the influence of likability the PD's payoff matrix was adjusted as compared to experiment 1 (see S1 Table). We argued that the likability manipulation would be more successful if a discrepancy between the participant's and confederate's decision (i.e., defect versus cooperate) altered points more strongly than experiment 1. In order to further strengthen the participant's focus on the likability manipulation during the experiment, participants were explicitly told about and received physical money rather than points after the prisoner's dilemma. Lastly, the participants played against four instead of six agents (see Table 2) with the goal to simplify and thus highlight the likability and functionality manipulations for the participants. This time no agents were assigned a neutral condition, allowing us to perform a two-way repeated measures ANOVA with *functionality* and *likability* as independent variables and *RT coupling* as dependent variable.

### 3.3 Results & discussion

**3.3.1 Response accuracies.** Similar to experiment 1, the percentage of late (M = 4.9%, SD = 3.1%) and incorrect trials (M = 4.5%, SD = 3.9%) did not differ across agent types (Late: $F(3, 123) = 2.10$, $p = 0.104$, $\eta_p^2 = 0.05$; Incorrect: $F(3, 123) = 1.14$, $p = 0.338$, $\eta_p^2 = 0.03$).

**3.3.2 RT coupling between agents and participants.** Like in experiment 1, reaction times of the participants changed as a function of the reaction times of the agents (See S1B Fig; $t(41) = 17.55$, $p < .001$, $d = 2.71$; Slope of linear fit: M = 0.26ms, SD = 0.09ms; Average correlation: M = .19, SD = .06), confirming that participants were also affected by the actions of the agents in this experiment. The strength of the relation between these reaction times (Fig 2C; for alternative plot, see S2B Fig) was affected by functionality ($F(1, 41) = 25.35$, $p < .001$, $\eta_p^2 = 0.40$) but not by likability ($F(1, 41) = 0.61$, $p = .439$, $\eta_p^2 = 0.01$; no interaction: $F(1, 41) = 0.01$, $p =$

**Table 2. Behavioral characteristics per agent in experiment 2.**

| Agent type | Prisoner's dilemma game *Percent cooperative trials* | Color action game *Percent correct choices* |
|---|---|---|
| Likable/Functional | 80% | 80% |
| Likable/Dysfunctional | 80% | 20% |
| Dislikable/Functional | 20% | 80% |
| Dislikable/Dysfunctional | 20% | 20% |

.934, $\eta_p^2 < 0.01$). The difference in RT coupling between functional and dysfunctional agents was significantly bigger than the difference between likable and dislikable agents ($t$(41) = 3.90, $p < .001$, $d = 0.60$). In line with the results of experiment 1, we conclude that (1) reaction times of functional agents affected the reaction times of participants more strongly than reaction times of dysfunctional agents, (2) likable agents did *not* have this effect as compared to dislikable agents, and (3) the effect of functionality on RT coupling was bigger than that of likability.

**3.3.3 Subjective awareness of agent manipulations.** The examination of the agent's likability on monetary rewards ($t$(41) = 22.37, $p < .001$, $d = 3.45$) and subjective ratings (Fig 2D; for statistics, see S5 and S6 Tables) confirmed that the PD game successfully manipulated how likable agents were perceived, even to a larger degree than the effect of the functionality manipulation had on action correctness during the CA game ($t$(41) = 5.70, $p < .001$, $d = 0.88$).

**3.3.4 Possible alternative explanation for a lacking effect of likability.** One not yet considered aspect of experiment 1 and 2 is that the likability manipulation was separated in time from the CA game during which the degree of perception-action pairing was measured. As it is known that an increase in time between the agent's and participant's actions results in weaker pairing [38], the effect of likability might have decreased as time between PD and CA game tasks passed. This means that the likability manipulation was more distal to the RT coupling measurement than the functionality manipulation. Another aspect that was not yet considered in experiment 1 and 2 is related to the possibility that incompatible actions by dysfunctional agents may have slowed down reaction times of participants (Brass et al., 2000), and, as such, confounded the correlations. Although it is unlikely that this potential confound weakens correlations because it would enlarge the range of reaction times and increase the probability of finding stronger correlations, we still wanted to exclude it as a potential mediator in experiment 3. To make the functionality manipulation as distant from the CA game as the PD game and to ensure that action compatibility did not affect reaction times, we preceded the CA game with a separate, third task in experiment 3 specifically for the functionality manipulation. The task consisted of a quiz with challenging questions on varying topics. The functional agents answered most of these questions right while dysfunctional agents answered most of them wrong. The order of the quiz and PD game was counterbalanced to prevent time effects (bottom panel in Fig 1B), and the error rate of the agents during the CA game (and thus the degree of performing functional actions) was equalized to an intermediate level across all agents.

# 4. Experiment 3

## 4.1 Introduction

Experiment 3 builds on experiment 2, with the aim to investigate if the effect of functionality still persists if the manipulation of functionality is separated from the perception-action pairing task.

## 4.2 Methods

All methodological aspects of experiment 3 were similar to experiment 2, except for the group of participants and experimental design. The group of participants in the third experiment consisted of 42 people (M = 22.21, SD = 2.68, range: 18–33, thirty women, six left-handed). To manipulate functionality separately from the CA game, we added a quiz game to the PD game before each sequence of CA games. Participants had to answer a total of 20 questions per agent (1 question per trial; bottom panel in Fig 1B). We collected the total of eighty questions for all 4 agents from several quiz websites. Because we now manipulated functionality in the quiz game, the color action game's only function was to measure the degree of RT coupling by

proxy of reaction time correlations. Therefore, all the agents scored on average during the CA game by setting the number of congruent trials at 50 percent (40/80 trials). Functionality now refers to the percentage correctly answered questions by the agents during the quiz-game. A functional agent answered 80 percent of the questions correct, a nonfunctional agent answered 50 percent of the questions correct.

## 4.3 Results & discussion

**4.3.1 Response accuracies.** The percentage of late (M = 5.1%, SD = 2.5%) and incorrect trials (M = 4.8%, SD = 5.4%) did not differ across agent types (Late: $F(3, 123) = 0.13$, $p = 0.945$, $\eta_p^2 < 0.01$; Incorrect: $F(3, 123) = 0.99$, $p = 0.400$, $\eta_p^2 = 0.02$).

**4.3.2 RT coupling between agents and participants.** Experiment 3 produced similar RT coupling results as experiment 1 and 2. Reaction times of the participants changed as a function of the reaction times of the agents (S1C Fig; Binned RT: $t(41) = 15.75$, $p < .001$, $d = 2.43$; Slope of linear fit: M = 0.24ms, SD = 0.14ms; Average correlation: M = .20, SD = .08). This form of perception-action coupling (Fig 2E; for an alternative plot, see S2C Fig) was affected by functionality ($F(1.1, 44.5) = 4.33$, $p = .040$, $\eta_p^2 = 0.10$) but not by likability ($F(1.1, 44.5) = 0.03$, $p = .889$, $\eta_p^2 < 0.01$; no interaction: $F(1.1, 44.5) = 2.64$, $p = .112$, $\eta_p^2 = 0.06$). In line with previous research [38], the explorative creation of a more distal functionality manipulation by means of the quiz game significantly weakened its effect on RT coupling by a factor 3 (experiment 2 versus experiment 3: $t(82) = 2.42$, $p = .018$, $d = 0.53$). We found no evidence that the effect of functionality on RT coupling was stronger than the effect of likability ($t(41) = 1.52$, $p = .137$, $d = 0.23$). We can thus conclude again that functional agents affected the actions of participants more strongly than dysfunctional agents, but likable agents did not affect actions more than dislikable agents.

**4.3.3 Subjective awareness of agent manipulations.** The effect of agent's likability on monetary rewards ($t(41) = 26.28$, $p < .001$, $d = 4.06$) and subjective ratings (Fig 2F; for statistics, see S7 and S8 Tables) again confirmed that the prisoner's dilemma game successfully manipulated the perceived likability. Importantly, the quiz game also successfully manipulated the perceived functionality (Positive: M = 65.06, SD = 8.63; Negative: M = 55.65, SD = 7.83; $t(41) = 5.59$, $p < 0.001$, $d = 0.86$). Thus, although participants were aware of the differences in likability and functionality across agents, only the functionality manipulation affected RT coupling.

## 5. General discussion

In three experiments, we investigated to what degree an agent's behavior, as to making correct (functionality) and cooperative decisions (likability), affected a participants' tendency to adjust their own reaction times to the agents' reaction times in a joint stimulus-response task. Across all experiments and task variations we observed significant correlations between the agents' and participants' reaction times, which were stronger for functionally behaving than nonfunctionally behaving agents. The correlations likely reflect the synchronization of reaction times, a rather specific form of (imitative) perception-action coupling.

In our experiments the likability of an agent had no effect on RT coupling. This may seem at odds with the well-established imitation and joint action literature showing effects of likability of agents on behavioral resonance of actions across space and time [12–15, 19, 20, 22, 23]. The effect of likability seems robust as it even facilitates imitation and synchronization of complex or uncommon behaviors, such as facial expressions [45, 55], rocking a chair [56], and the swinging of a pendulum during conversation [57]. It is important to note that such effects of likability are often investigated in physical rather than virtual, computer environments [58].

However, research using virtual environments and RT tasks suggests that specifically the effects of the social presence of a person does not necessarily explain perception-action coupling [33]. Instead, it is the a priori knowledge about an actor's presence that affects reaction times [59]. Such knowledge may set the action schemes of observers, like the possibility that, as agents performed more and more correct actions, participants may have gradually strengthened their focus of attention for the actions of agents [4, 31]. It is thus possible that not social contexts but factors that control attention affect perception-action coupling in specifically reaction time and stimulus-response tasks.

One alternative explanation as to why we do not find an effect of likability could be that action functionality plays such a dominant role in virtual as compared to physically interactive tasks to a degree that it suppresses potential subordinate effects of likability on reaction times. This suppression could be the result of an interaction between different cortical mechanisms involved in automatic and intentional imitation. Several cortical networks have been discovered to play a role in imitative perception-action pairing, including the mirror-neuron network [60, 61]. The network responsible for intentional and goal-directed imitation [62] could control automatic coupling of the timing of actions [63], depending on contextual factors such as an agent's likability. Interestingly, several brain areas in these perception and action networks overlap with brain areas involved in the evaluation of fairness and likability of others [64, 65], suggesting that a suppressive interaction between the evaluative processes of action purpose and action likability. Indeed, Etzel and colleagues [65] manipulated fairness with the same prisoner's dilemma game as used in the current study and found that the activity of brain regions involved in perception and action are not modulated by the fairness and thus likability of observed agents. This finding thus concurs with the absence of a likability effect in the current experiment.

One limitation of the current study is that we investigated a rather narrow form of (imitative) perception-action coupling, namely time coupling, reserves us from generalizing the current findings to other forms of imitation and mimicry. Also, the complexity of sequential actions in interactive dyads have not been explored, with fractal scaling and recurrence quantification as potentially insightful analyses for how synchronization of reaction times may dynamically vary over time [66–69]. Another limitation of the current study is a weaker effect of functionality and the lacking interaction between functionality and likability in experiment 3. This weaker effect is most likely the result of the change in the manipulation type (i.e., the quiz) and the increased duration between the agent behavior manipulation and the reaction time coupling test. We argue that varying the correctness of an agent's answers during a quiz might not be as effective for functionality as the cooperation trade-off manipulation in the prisoner's dilemma game was for likability. While we took several actions to improve the likability manipulation across experiments, and we still consistently observed no effects of likability, we expect that the improvement of the functionality manipulation in future research should lead to functionality effects that are significantly stronger than the nonsignificant effects of likability.

One not yet considered factor that may have affected the results is the believability of the physical presence of human agents [30, 31]. At the start of the experiment, participants were led to believe they would be playing with agents via a real time video stream. During the debriefing, the experimenters occasionally asked a participant if they believed the existence and authenticity of the different agents. A majority of the participants indicated that they were not entirely sure whether they played with human agents via a live real time or prerecorded video stream. Despite this uncertainty, the answers to subjective questionnaires showed that participants were conscious of the diverging behavioral profiles of the agents. They reported that the agents' behaviors affected their feelings towards them (e.g. feelings of annoyance or even anger towards dislikable agents), making the social cooperativeness manipulation successful. Also, we ensured that full attention was paid to the actions of the agents by

superimposing the color cue on top of the hand performing the action. As attention is likely the mediating factor of effects of believability of human presence (Dolk, et al., 2014; Sellaro, et al., 2013), we do not expect that the artificial context and believability explains the lacking effects of likability in the current study. Thus, despite the artificial environment used in this study, despite the possibility that some participants may have not fully believed that they were playing against a real person, and despite the unanswered question whether current results generalize to physically interactive settings, the fact that functionality but not likability affected the actions of the observer in a computerized stimulus-response task is a novel finding and the first step in comparing functional accounts of perception-action coupling.

We end with the question asked in the introduction: which factors most significantly evoke RT coupling between people? We are confident that, at least in virtual stimulus-response compatibility tasks and for temporal aspects of behavior, the presence of action errors is one of the main drivers of RT *de*coupling. This explains why the degree of perception-action coupling can modulate so strongly from trial to trial depending on the usefulness of precedingly observed actions [38]. It is tempting to propose that the function of RT coupling–and perhaps also of other forms of perception-action coupling–is strongly driven by the possibility of taking advantage of learning [6–11] and dynamic dyadic action coordination in joint tasks [70, 71]. The role of bonding, however, remains to be confirmed in future studies that may adopt the current design and incorporate more realistic, physical settings. We hope that more research will shed light on the cause of the intriguing underlying mechanisms that allow us to copy useful actions from others without too much creative, internal effort that is normally required for self-produced actions.

## Supporting information

**S1 Fig. Reaction times of participants as a function of reaction times of agents per experiment.** Linear regression fits to reaction times per trial of participants (y-axis) as a function of reaction times of sagents (x-axis) per experiment (A-C). Note that the distribution of reaction times of agents varied across individuals due to random sampling of reaction times.
(DOCX)

**S2 Fig. Reaction times per agent type per experiment.** Mean and standard errors across participants of correlations of reaction times between participants and agents (y-axis) per agent type (x-axis) and per experiment (A-C). Func = Functional (correct choices), Dysfunc = Dysfunctional (incorrect choices), Neu = Neutral, Like = Likable (cooperative), Dislike = Dislikable (uncooperative).
(DOCX)

**S1 Video. Example of a single trial in which the participant plays a prisoner's dilemma game and a color action game.** The screenshot shows the first screen of a block during which a new agent is introduced to the participant.
(MOV)

**S1 Table. The pay-off matrix of the PD in experiment 1 and experiment 2 & 3.** Based on their choice to defect or cooperate, the participant and agent could earn or lose points. In experiment 2 & 3 participants could earn or lose money (numbers are in cents rather than points).
(DOCX)

**S2 Table. Questions to evaluate to what degree the behavioral manipulations of the agents were noticed by the participants.** The questions had to be answered on a continuous

dimension scale of 0 to 100.
(DOCX)

**S3 Table. Experiment 1 one-way ANOVA results (F-statistic, p-value) on subjective ratings, with the predicting factors of agent type per rating type.**
(DOCX)

**S4 Table. Experiment 1 post-hoc t-test results (t-value, p-value) on subjective ratings compared across agent types (dof = 29; number represent comma-separated t-statistics and p-value).**
(DOCX)

**S5 Table. Experiment 2 two-way repeated measures ANOVA results (F-statistic, p-value) on subjective ratings with agent functionality (dof = 1, 41) and agent likability as predictors (dof = 0.9, 36.4).**
(DOCX)

**S6 Table. Experiment 2 post-hoc t-test results (t-value, p-value) on subjective ratings compared across agent types (def = 41).** Abbreviations: L+ F+: likable, functional; L+ F-: likable, dysfunctional; L- F+: dislikable, functional; L- F-: dislikable, dysfunctional.
(DOCX)

**S7 Table. Experiment 3 two-way repeated measures ANOVA results (F-statistic, p-value) on subjective ratings with agent functionality and agent likability as predictors (dof = 1, 41).**
(DOCX)

**S8 Table. Experiment 3 post-hoc t-test results (t-value, p-value) on subjective ratings compared across agent types (dof = 41).** Abbreviations: L+ F+: likable, functional; L+ F-: likable, dysfunctional; L- F+: dislikable, functional; L- F-: dislikable, dysfunctional.
(DOCX)

## Author Contributions

**Conceptualization:** Zoe Schielen, Julia Verhaegh, Chris Dijkerman, Marnix Naber.

**Data curation:** Zoe Schielen, Julia Verhaegh, Marnix Naber.

**Formal analysis:** Marnix Naber.

**Investigation:** Zoe Schielen, Julia Verhaegh, Marnix Naber.

**Methodology:** Zoe Schielen, Julia Verhaegh, Marnix Naber.

**Project administration:** Marnix Naber.

**Resources:** Chris Dijkerman, Marnix Naber.

**Software:** Marnix Naber.

**Supervision:** Chris Dijkerman, Marnix Naber.

**Validation:** Zoe Schielen, Julia Verhaegh, Marnix Naber.

**Visualization:** Marnix Naber.

**Writing – original draft:** Marnix Naber.

**Writing – review & editing:** Zoe Schielen, Julia Verhaegh, Chris Dijkerman, Marnix Naber.

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
