## [Decision Letter · Decision Letter 0]

28 Jul 2021

PONE-D-20-38766

Perception-action pairing in a stimulus-response compatibility task: a matter of social bonding or learning?

PLOS ONE

Dear Dr. Naber,

Thank you for submitting your work to PLOS ONE, and I apologize for the length of time for the review. The stresses of the global pandemic have caused significant delays in the peer review process, as it has affected many other parts of life these days.

Two experts in related areas have reviewed the manuscript, and I have evaluated it myself. While all of us find your research question to be interesting, there are substantial issues with the manuscript in its current form, including theoretical grounding, methodological questions, and empirical limitations. The reviewers and I agree that this direction could have promise, but considerable work would need to be done before the work is ready for publication. As a result, I am making a determination of Major Revision.

Both reviewers have provided very detailed commentary and suggestions on the current manuscript. I will not repeat all of their points here, but I encourage you to attend to their points carefully if you decide to prepare and submit a revision. I will draw attention to perhaps the most important theoretical point raised by both reviewers: the potential of more domain-general, non-social dynamics in driving the effects. Addressing this point will be critical in a revision, especially given the limitations of the current empirical evidence and the importance of the social context to the current theoretical arguments. I am unsure whether it will be possible to adequately address these concerns without additional data collection.

We look forward to receiving your revised manuscript.

Kind regards,

Dr. Alexandra Paxton

Academic Editor

PLOS ONE

Journal Requirements:

3. PLOS requires an ORCID iD for the corresponding author in Editorial Manager on papers submitted after December 6th, 2016. Please ensure that you have an ORCID iD and that it is validated in Editorial Manager. To do this, go to ‘Update my Information’ (in the upper left-hand corner of the main menu), and click on the Fetch/Validate link next to the ORCID field. This will take you to the ORCID site and allow you to create a new iD or authenticate a pre-existing iD in Editorial Manager. Please see the following video for instructions on linking an ORCID iD to your Editorial Manager account: https://www.youtube.com/watch?v=_xcclfuvtxQ.

5. We note that Figure 1A includes an image of a participant in the study. 

As per the PLOS ONE policy (http://journals.plos.org/plosone/s/submission-guidelines#loc-human-subjects-research) on papers that include identifying, or potentially identifying, information, the individual or parent(s)/guardian(s) must be informed of the terms of the PLOS open-access (CC-BY) license and provide specific permission for publication of these details under the terms of this license. Please download the Consent Form for Publication in a PLOS Journal (http://journals.plos.org/plosone/s/file?id=8ce6/plos-consent-form-english.pdf). The signed consent form should not be submitted with the manuscript, but should be securely filed in the individual's case notes. Please amend the methods section and ethics statement of the manuscript to explicitly state that the patient/participant has provided consent for publication: “The individual in this manuscript has given written informed consent (as outlined in PLOS consent form) to publish these case details”. 

Reviewers' comments:

Reviewer's Responses to Questions

**Comments to the Author**

1. Is the manuscript technically sound, and do the data support the conclusions?

Reviewer #1: Partly

Reviewer #2: No

2. Has the statistical analysis been performed appropriately and rigorously? 

Reviewer #1: Yes

Reviewer #2: I Don't Know

3. Have the authors made all data underlying the findings in their manuscript fully available?

Reviewer #1: No

Reviewer #2: No

4. Is the manuscript presented in an intelligible fashion and written in standard English?

Reviewer #1: Yes

Reviewer #2: No

5. Review Comments to the Author

Reviewer #1: Summary

The authors investigated whether coupling strength between individuals (measured via the variation in correlation in response times) is impacted by the perceived likability of the partner, or their perceived competence in unintentional coordination. Manipulations to like-ability was done using the prisoner’s dilemma game, and manipulations to the competency of the partner (or “functionality”) was manipulated by their congruency of responses to the CA game (Experiments 1 and 2) or their competency to answer questions on a quiz (Experiment 3). To confirm the manipulations had the desired effect, participants also answered a set of questions rating the agents’ likeability and competency (functionality).

Across the three experiments, the researchers found that variation in the competence of the agent in all three experiments impacted the degree of coupling in response times between the participant and the agent in the CA game. Variation in likeability did not change the correlation. Thus, the researchers demonstrate that making the competency of the partner salient impacts the degree of perceptual-motor coupling to that partner.

General Comments/Questions

The authors have drawn from speeded response paradigms to motivate their research study. There is a line of research from researchers in the ‘coordination dynamics’ literature that has investigated the role of pattern stability and social factors on interpersonal coordination whose inclusion would benefit this paper. In particular, the work of Lynden Miles (University of Western Australia) would be of relevance here. Miles has investigated the effects of rapport and group membership, for example, on the degree of coupling between people during rhythmic coordination tasks. In their work, social variables has an influence on the amount of coordination that is observed between people.

I agree with the authors that despite possible limitations in their study, the fact that manipulations to competence, but not likability, had an impact on response time correlations is a novel finding. These findings support the view that coupling between one’s own action and what is observed is influenced related to their predictability.

One major challenge I have with the presentation of the study and the interpretation of the findings is how quickly the researchers are to conclude that the findings are tied to social imitation and mimicry, as opposed to a potentially non-social explanation. This cannot be fully determined as there was no “non-social” version of the task presented to participants. For example, would the researchers expect the same findings in re: to functionality if there was no “second person” on the experiment? For example, would the presentation of divergent answers in the quiz produce similar findings in Experiment 3? The conclusion I draw from the Experiments is that predictability of the task environment modulates attention by participants to complete the task. The researchers should acknowledge that this is a possibility and a limitation to their work, or provide a stronger argument why this is a social effect

Specific Comments

The title is loaded with terms I think is inappropriate given the task and findings. “Likeability” might be more appropriate than “Social Bonding” and I would argue that “learning” is an inappropriate word here and something along the lines of “modulation of attention” would be more appropriate.

For the color action (CA) game, were participants told that the videos that played were from the participant they were completing the task with, or pre-recorded videos? If participants are told they are pre-recorded – are they told they are of their partner?

For the CA game in Method, clarify what is a congruent and incongruent action.

Lines 284-286. I’m confused what this statistical test is compared to the t-tests in the preceding sentences (lines 280-282). Also, results for Exp. 2 and 3 were generally easy to follow but was a little more of a challenge for Exp. 1. May I suggest using sub-headings that provide a predictable structure for all 3 experiments.

Line 358. No Text under “Introduction” subheading. Unsure if this was intentional

It is unclear in the text what was the nature of the pre-recorded videos in the CA game for Experiment 3. Were all the videos congruent-only? (I assumed it is).

In re: Discussion. Similar to my concerns with the title, I think potentially loaded terms are being used to explain the findings. For example attributing “trustworthiness” to the second person to explain the coupling correlations might be too much attribution for something that may be due to modulation of attention due to the predictability of responses in the task.

Related to my concerns above, my thoughts are motivated by similar concerns raised by Thomas Dolk’s work on referential coding in the joint simon task – where, certain “social” effects can be explained non-socially. I have not read the Dolk 2014 article the researchers cite, but I’m curious if that makes similar conclusions. This may also explain why the likability manipulation did not have an impact on the findings – because the explanation of the findings may be non-social.

Reviewer #2: Content

The paper’s aim was to find out, as Author write: “To what degree the functionality and likability of agents shape reaction time synchronization”. The synchronization was measured as a correlation of reaction times in a congruency color task (“the same game”) played by an alleged confederate “several milliseconds in advance”.

Agents varied in likeability and “functionality”, which was evoked by manipulating their cooperativeness in prisoner’s dilemma and correctness of choices in the “preceding” games. The correlations were affected more by “functionality” of the agents than their likeability. Two more experiments were conducted to check if a stronger likeability manipulation would affect the correlations (it did not), and to manipulate the skillfulness (“functionality”) of the agents in a different way than with the same task on which the correlations were measured.

Author conclude that a stronger correlation of the RTs with “functional” agents means that it is the “potential to benefit of learning from others” that drives perception and action coupling.

Evaluation

The question posed by the Authors (if the timings of actions become more similar when the partner is perceived as cooperative and competent) is potentially interesting. The manipulations seemed to have the desired effect as shown by the ratings. A potentially interesting result is a weak but significant correlation of the reaction times in general.

However, I have strong reservations as to the other aspects of the paper. I had difficulty to ascertain from the descriptions if the design of the study indeed allowed to address the research questions, and if the questions were posed clearly on the theoretical level.

Theory and formulating research question and hypotheses:

Some of the terms in the paper are used in a non-standard way. For example, the first sentence of the abstract says: “Being influenced by others’ behavior is a deep-rooted mechanism.” Is this really a mechanism, or we are rather seeking mechanisms underlying such a phenomenon?

I am not sure why the phenomenon is called “perception-action” pairing. Isn’t it rather matching of the perceived timings of actions of the collaborator with the timings of own actions? I will assume the Author’s expression is a shortcut for this. But it is still not clear to me if the Authors will research only the general time-coupling or synchronization of behaviours or the timings of the imitative behaviours only. In other parts of the paper Authors write about “imitative perception-action timings” or simply “imitation” (in the Introduction).

Evoking learning seems to point to this narrower interpretation, I am not sure about social bonding. Both learning and bonding are not the only reason why people would synchronize their imitative or complementary actions – the most obvious reason is cooperation in dyads and groups for which basic time-synchrony seems to be important as shown in many studies (e.g. research by Carol Fowler, Kerry Marsh, Michael Richardson, Kevin Shockley, Rick Dale or Alex Paxton). This larger context, in which imitation for learning and bonding could be presented, is ignored in the paper. It is a pity because learning takes place mostly while doing things WITH others (for which timing is also important) and not observing others and imitating the shape and timing of their moves. Perhaps such a larger context would help the reader understand better the motivations for posing the the research question: for this reviewer it was not clear, why is it important to know if our coupling of action timings is stronger because of the competence or liking of the others? Why are the two contrasted with each other, while they could well be both?

Empirical level:

I am not sure I fully understood the design and the analyses.

The color action game was used, which had two goals: to “manipulate whether the agent played good” and “measure the degree of similarity in reaction times”, which, if I understand correctly, put the participant in a task of – sometimes – imitating the agent and sometimes suppressing the imitation and going for the right color (if the agent in the video pressed the wrong button). This occurred with various frequency for various agents (which are called functional or dysfunctional because of that).

So if someone wanted to measure the timings correlation in imitative movements, the RTs should be taken only from the correct trials of the agent and participant. Although it is not clearly written in the paper, if I understand correctly the rationale behind Experiment 3, and the lack of addressing the problem of unequal RT sample sizes (80%correct for some agents and 20% correct for the others) – this was not the case. Thus the RTs were most likely measured across all the trials (congruent – when the movements of agents and participants matched) and incongruent (when they mismatched). If this was the case, then could the results be due to the participants being more often primed for the correct action for those agents who were more often correct, while when they were more often not correct, the participants had to inhibit the whatever desire to imitate (if they had any) and perform a different action. In the case of agent’s movements which started before the cue for the participants (this is how -200ms could be interpreted), this additionally gave a hint to the direction, which probably facilitated the movement of the participant resulting also in fast RTs, which for the competent agents could be taken as reliable. In short: Seeing an incongruent movement the participants had to disregard or suppress the primed reaction observed in the confederate, which could take variable time depending on the strategy they used. Seeing a congruent movement, they saw it as a cue for their own.

If I am reading this wrong, and if only the “imitative” (though it is perhaps hard to call them such in a task requiring exactly the same movement) trials were taken into account then it should be explained how the problem of unequal trial numbers were dealt with. And if so – why the Exp 3 was performed (one of its aim was to eliminate the congruency bias).

The task of the reader is made more difficult by the fact that the detailed instructions and full information on what the participants heard and saw is missing. How exactly was the agent introduced? As a collaborator, as an opponent? How was it explained that the video of the other person was played to the participants in the CA task? This is important as this is how participant’s action is coupled to that of the “partner” – this gives a larger setting for the task.

Other instructions and informations are also missing: How did the Authors ask about a person’s “functionality” (BTW I do not think labeling agents functional or dysfunctional on the basis of CA game is fortunate). Were the points in PD accumulated and visible for the participants? What was the feedback for, etc, etc….?

Other comments:

Prisoners dilemma needs often several rounds in order to establish a partner as cooperative or competitive. Here one PD game was followed by 4 CA games - could one measure (only in congruent trials what happened on each block depending on the PD behaviour? Also, a series of cooperations or defeats should influence the judgment on cooperativity in a long run (throughout the 20 blocks). The Authors performed some split analyses (without clear motivations and hypotheses) but not this one.

I did not understand the idea that participants might have waited when the answer was congruent. Later in the paper Authors propose that perhaps they waited at the incongruent trials (lines 404-405, this time perhaps legitimately). But it is not clear why in the first case this should have led to a greater correlation of the timings and in the second not.

Authors wrote that it was possible that the participants did not believe the other has indeed been present and wasn’t an avatar – why this was not checked at the end by simply asking the participants? Also: wasn’t it highly suspicious if an agent erred on 80% of the trials in such a simple task?

I am at a loss why experiments 2 and 3 were conducted if the experiment one showed that the “likeability” was successfully effectuated. Also: Could the analyses be done on likeability itself and not only on the propensity to defeat and cooperate, if it was the likeability that was hypothesized to influence the timing matching and it was measured? Experiment 3, in turn, used a strikingly different task for measuring “functionality”, tapping on completely different skills. What is the basis for claiming that competence in one task will affect the perceived competence in the other?

Minor remarks:

• I am confused why joint Simon task is claimed to be less interactive and involve “two person’s perform unrelated and time separated actions” (line 68)

• In the study where cooperativeness, social bonding etc. are studied, it would be good to balance the participants group for gender or – better – included as a factor.

• What does it mean that something is “sequentially intermixed” (line 137)

• The agents were 5 females and 1 male what was the reason for that?

• 4% of the answers were errors and 5% were late – why were they not described qualitatively? Were they congruent with the wrong button presses by the “confederates? Did they occur only on incongruent trials?

• I have several reservations regarding the statistical analyses and reporting but they have no consequences in the face of the more significant doubts above.

In summary I do not think this work managed to show that "functionality" affects timing sychronization and have to recommend rejection.

6. PLOS authors have the option to publish the peer review history of their article (what does this mean?). If published, this will include your full peer review and any attached files.

Reviewer #1: No

Reviewer #2: No

---

## [Author Response · Author response to Decision Letter 0]

19 Oct 2021

See attached response to reviewers

---

## [Decision Letter · Decision Letter 1]

20 Apr 2022

PONE-D-20-38766R1Reaction time coupling in a joint stimulus-response task: a matter of functional actions or likable agents?PLOS ONE

Dear Dr. Naber,

Thank you for submitting your manuscript to PLOS ONE. After careful consideration, we feel that it has merit but does not fully meet PLOS ONE’s publication criteria as it currently stands. Therefore, we invite you to submit a revised version of the manuscript that addresses the points raised during the review process.

The manuscript has been further evaluated by two reviewers, and their comments are available below.

While almost ready for publication, one of the reviewers notes the need for additional clarification in the reporting of the statistical procedures and interpretation of the results.

Could you please revise the manuscript to carefully address the concerns raised?

We look forward to receiving your revised manuscript.

Kind regards,

Avanti Dey, PhD

Staff Editor

PLOS ONE

Reviewers' comments:

Reviewer's Responses to Questions

**Comments to the Author**

1. If the authors have adequately addressed your comments raised in a previous round of review and you feel that this manuscript is now acceptable for publication, you may indicate that here to bypass the “Comments to the Author” section, enter your conflict of interest statement in the “Confidential to Editor” section, and submit your "Accept" recommendation.

Reviewer #1: (No Response)

Reviewer #3: All comments have been addressed

2. Is the manuscript technically sound, and do the data support the conclusions?

Reviewer #1: Partly

Reviewer #3: Yes

3. Has the statistical analysis been performed appropriately and rigorously? 

Reviewer #1: Yes

Reviewer #3: Yes

4. Have the authors made all data underlying the findings in their manuscript fully available?

Reviewer #1: (No Response)

Reviewer #3: Yes

5. Is the manuscript presented in an intelligible fashion and written in standard English?

Reviewer #1: Yes

Reviewer #3: Yes

6. Review Comments to the Author

Reviewer #1: I would like to thank the authors for taking my comments and suggestions seriously. I think the edited changes make the document more understandable (for example, in the Method). The authors incorporated a wider literature review, and also provide additional acknowledgement regarding non-social explanations for the findings (with a nod to Dolk’s 2014 review article – although I am still confused by the interpretation of Dolk – see below).

Although the authors have made changes to reduce to provocative nature of the title and the text, I think further work needs to be done to acknowledge the vast research that has looked at coupling/synchronisation in all of its complexities. For example, the introduction and discussion give the impression that this study would definitely determine whether coupling (in RT contexts) has functional or social bonding benefits. For example in the Discussion “Although it is tempting to propose” with what is written following that is too strong. The results presented in this manuscript, although novel, is not convincing enough to change my view regarding the role of coupling/synchronisation in perceptual-motor tasks.

The authors introduce the multitude of benefits for synchronisation with the inclusion of additional references in this revision. I think it is enough to justify the study. However, I think the interpretation of the results should be constrained to this particular task context without language that gives the impression that the role of coupling/synchronisation is X over Y - where the literature suggests that X and Y are both important.

Section comments:

Abstract:

1. “Here we employ a joint-action task” – that sentence should be rewritten to be clearer on what aspect was “joint” (the prisoner dilemma game. I would not call the color game a joint action task).

2. “Functionality and likeability – be more specific. “Functionality = predictability that the agent will make the same response. “Likeability” – cooperation in prisoners dilemma game.

3. The last sentence “these findings suggest” include “are more likely to adopt”.

Introduction:

I am confused by the interpretation of the Dolk et al 2014 review article. Dolk et al also review research that demonstrate that performing a “joint” task next to a metronome or a cat can also induce these effects. How do we make sense of these results and the results by Tsao 2008? I think a couple more sentences are needed that reviews these lines of research (and anything more contemporary) to justify the use of the artificial agent setup to pursue research question.

Line 110 “the chance to team up” – in the context of this experiment, there is no chance to “team up” during the reaction time task so how can likeability influence behavior in this context?

Results:

1.) Please add effect sizes to your tests.

2.) Was there an official debriefing protocol that was followed to determine whether participants believed they were interacting with a human? What % of participants were/were not convinced?

Discussion:

Last paragraph. “Although it is tempting to propose” This last sentence needs to be reworked. The role of synchronisation is multi-faceted. I would simplify is that the role of synchronisation is for coordination of minds. This has a function role (to help with timing of actions, to be “in sync” on how to solve a task) as well as a bonding/social role (to build rapport, to empathise). I would argue that the results in this experiment, given that there is no true interactive context, are not strong enough to “tilt” the scale towards synchrony having functional over social benefits.

I have no issues regarding the results themselves, but I would advise the authors to really temper down claims of generalisation to coupling/synchronisation generally.

Article to consider:

A recent article (Wohltjen & Wheatley 2021 “Eye contact marks the rise and fall of shared attention in coversation” would be a useful resource for the authors which investigates coordination in a naturalistic context. In particular, the discussion section has a lot of good articles that might be of value related to the social and functional aspects of synchrony in human interaction.

Another article that might be useful for you (although less clear): Dotov et al 2019 “The Role of Interaction and Predictability in the Spontaneous Entrainment of Movement.” The description of the abstract seems the most relevant, but the authors investigated participants participants to synchronise to auditory signals that were either interactive or predictable.

Reviewer #3: Review of PONE-D-20-38766R1, Functionality rather than likability drives reaction time coupling.

This joint-action response time study is motivated by the social synchronization literature. Shared goals and a common social context often result in coordination and synchronization in individual's movements, gestures, and communication. This article tested for influences originating from both the functionality and the likability of virtual partners. The results indicate that the participants response times became correlated with those of the virtual agents. On the other hand, social likability did not appear to influence the outcomes. The authors conclude that perception-action synchronization is more important for the SR compatibility task.

The authors express a bit of surprise that likability did not impact RT performance. However, I don't find it too surprising. Most of those published studies involved interpersonal interactions that evolved over some length of time (e.g., conversations, games). The synchronization measure typically emphasized events on much shorter time scales (gestures, movements, utterances). Most of these situations require adjustments just to maintain the social interaction.

By contrast, the likability manipulation and the RT task happen on relatively fast and similar time scales. And their nature varies as a function of trial. I agree that a lack of functionality is the crucial factor given the design. However, one must be mindful that all the activities involved in going to a lab, the consent process, and the interactions with competent and engaged experimenter is a compelling social context for participants. The authors are good experimenters, and my hunch is the participant's desire to please the researchers overrode the annoyances or distortions of the agents. Replace your well trained team with a pedantic, condescending, annoying and disinterested researcher for a couple days, and I guarantee your lab's error rates will explode, and your RT's will be a hot mess.

In my judgment, the manuscript meets or exceeds the 7 PLOS publication criteria. There may be some skepticism because the crux outcome constitutes a single pattern of correlation. However, it is replicated during the course of separate studies. The limitations in the discussion could be balanced a bit with more complete discussion of the synchronization hypothesis/outcome. In this regard, the authors may want to consult the literature on synchronization and fractal scaling in response time tasks. It is a better fit than the social sync literature, given the outcomes.

While too few data points are collected in the present task to allow the same analyses to be conducted, this literature has clearly established that response time performances are rooted in synchronization principles. 1/f noise is a stochastic form of multi-scale synchronization. There are many studies on this phenomenon. Adding noise to inter-trial intervals (ITIs) of response time tasks whitens the scaling, which is symptomatic of weaker coordination. Participants entrain to sinusoidal ITI fluctuations, and express frequency detuning, and mutual entrainment in dyadic temporal estimation performance. Sequential effects in two-alternative choice response time task were successfully captured with a discrete sine-circle map of the Haken Kelso Bunz (1985) bimanual coordination model. The participants index fingers behave as coupled oscillators, and the response durations and error rates are predicted, in surprising detail, by Farey tree ratios. Limited analyses of some of the sequential effect patterns (e.g., repetition) could be executed on the data in this manuscript. In the future, the authors should include explicitly dynamic manipulations of entrainment and synchronization, and collect many more trials. It is difficult to make a compelling case for something as dynamic as synchronization with a statistical toolset that was designed for static systems (e.g., t-tests, ANOVA, correlation etc.). Fractal analyses, recurrence quantification analysis and related statistical tools are helpful in this regard. My recommendation for publication is NOT contingent on these recommendations, but they might improve the manuscript, and inspire new studies.

7. PLOS authors have the option to publish the peer review history of their article (what does this mean?). If published, this will include your full peer review and any attached files.

Reviewer #1: No

Reviewer #3: No

---

## [Author Response · Author response to Decision Letter 1]

8 Jun 2022

See attached "response to reviewers" file

---

## [Editor Report · Decision Letter 2]

27 Jun 2022

Reaction time coupling in a joint stimulus-response task: a matter of functional actions or likable agents?

PONE-D-20-38766R2

Dear Dr. Naber,

We’re pleased to inform you that your manuscript has been judged scientifically suitable for publication and will be formally accepted for publication once it meets all outstanding technical requirements.

Kind regards,

John G. Holden

Guest Editor

PLOS ONE

Additional Editor Comments (optional):

Dear Authors, I am writing to notify you of my recommendation to accept your manuscript for publication in PLOS ONE. You've responded to several reviews and addressed the primary concerns of the reviewers. I would also like to disclose that I served previously as a reviewer on this manuscript. Overall, the manuscript is well balanced meets or exceeds the PLOS ONE criteria for publication. Congratulations!

Sincerely, John G. Holden
---

## [Editor Report · Acceptance letter]

1 Jul 2022

PONE-D-20-38766R2 

Reaction time coupling in a joint stimulus-response task: a matter of functional actions or likable agents? 

Dear Dr. Naber:

I'm pleased to inform you that your manuscript has been deemed suitable for publication in PLOS ONE. Congratulations! Your manuscript is now with our production department. 

Kind regards, 

on behalf of

Dr. John G. Holden 

Guest Editor

PLOS ONE